# GL-II-73, a Positive Allosteric Modulator of α5GABA_A_ Receptors, Reverses Dopamine System Dysfunction Associated with Pilocarpine-Induced Temporal Lobe Epilepsy

**DOI:** 10.3390/ijms241411588

**Published:** 2023-07-18

**Authors:** Alexandra M. McCoy, Thomas D. Prevot, Dishary Sharmin, James M. Cook, Etienne L. Sibille, Daniel J. Lodge

**Affiliations:** 1Department of Pharmacology and Center for Biomedical Neuroscience, UT Health San Antonio, San Antonio, TX 78229, USA; mccoya1@livemail.uthscsa.edu; 2South Texas Veterans Health Care System, Audie L. Murphy Division, San Antonio, TX 78229, USA; 3Campbell Family Mental Health Research Institute of CAMH, Toronto, ON M5S 2S1, Canada; thomas.prevot@camh.ca (T.D.P.); etienne.sibille@camh.ca (E.L.S.); 4Department of Psychiatry, University of Toronto, Toronto, ON M5T 1R8, Canada; 5Department of Chemistry and Biochemistry, University of Wisconsin-Milwaukee, Milwaukee, WI 53201, USA; dsharmin@uwm.edu (D.S.); capncook@uwm.edu (J.M.C.); 6Department of Pharmacology and Toxicology, University of Toronto, Toronto, ON M5T 1R8, Canada

**Keywords:** dopamine, temporal lobe epilepsy, electrophysiology

## Abstract

Although seizures are a hallmark feature of temporal lobe epilepsy (TLE), psychiatric comorbidities, including psychosis, are frequently associated with TLE and contribute to decreased quality of life. Currently, there are no defined therapeutic protocols to manage psychosis in TLE patients, as antipsychotic agents may induce epileptic seizures and are associated with severe side effects and pharmacokinetic and pharmacodynamic interactions with antiepileptic drugs. Thus, novel treatment strategies are necessary. Several lines of evidence suggest that hippocampal hyperactivity is central to the pathology of both TLE and psychosis; therefore, restoring hippocampal activity back to normal levels may be a novel therapeutic approach for treating psychosis in TLE. In rodent models, increased activity in the ventral hippocampus (vHipp) results in aberrant dopamine system function, which is thought to underlie symptoms of psychosis. Indeed, we have previously demonstrated that targeting α5-containing γ-aminobutyric acid receptors (α5GABA_A_Rs), an inhibitory receptor abundant in the hippocampus, with positive allosteric modulators (PAMs), can restore dopamine system function in rodent models displaying hippocampal hyperactivity. Thus, we posited that α5-PAMs may be beneficial in a model used to study TLE. Here, we demonstrate that pilocarpine-induced TLE is associated with increased VTA dopamine neuron activity, an effect that was completely reversed by intra-vHipp administration of GL-II-73, a selective α5-PAM. Further, pilocarpine did not alter the hippocampal α5GABA_A_R expression or synaptic localization that may affect the efficacy of α5-PAMs. Taken together, these results suggest augmenting α5GABA_A_R function as a novel therapeutic modality for the treatment of psychosis in TLE.

## 1. Introduction

Temporal lobe epilepsy (TLE) is a disorder characterized by the presence of spontaneous seizures, or uncontrolled neural activity, involving the temporal lobe, typically the hippocampal formation. Given that seizures are the hallmark feature of TLE, other neuropsychiatric comorbidities, such as psychosis, are often overlooked, despite the well-established relationship between TLE and psychotic symptoms [1,2,3,4]. TLE patients are 6–12 times more likely than the general population to suffer from psychosis [5,6]; however, the management of psychosis for patients with TLE is currently inadequate. Both first- and second-generation antipsychotics are associated with an increase in seizure incidence, as they can lower seizure thresholds [7,8]. Additionally, antipsychotics often produce intolerable side effects that lead to the discontinuation of medication [9], and have potential pharmacokinetic and pharmacodynamic interactions with antiepileptic drugs [7]. Thus, there are limited treatment options for the management of psychosis in patients with TLE. 

The hippocampus is a region that is central to the pathology of both TLE and psychosis, and is the area of the brain with the lowest seizure threshold [10,11]. Although psychosis is not present in all patients with TLE, they are 6–12 times more likely to develop psychosis compared to the general population [6], with psychosis risk being associated with numerous factors including early onset of epilepsy, a history of seizures, hippocampal sclerosis, and hippocampal abnormalities including cell loss in the CA1 region (for review, see [2]). It has been suggested that deficits in GABAergic signaling in the hippocampus contribute to some forms of epilepsy [12], perhaps due to epilepsy-associated interneuron death [13]. Interestingly, hippocampal abnormalities including increased activity at rest have been observed in psychosis patients, and are correlated with psychosis symptom severity [14,15,16,17,18]. Additionally, a loss of parvalbumin (PV)-positive interneurons in the hippocampus has been reported in post-mortem studies of psychosis patients [19] and in rodent models used to study psychosis [20]. Thus, a loss of inhibitory tone and subsequent hyperactivity in the hippocampus may be a point of pathological convergence for both TLE and psychosis. Indeed, our laboratory, and others, have shown that hyperactivity in the ventral hippocampus (vHipp) drives aberrant dopamine system function in rodents [21,22,23], with increased dopamine neuron activity being critical for generating symptoms of psychosis [24,25]. Importantly, preclinical studies have demonstrated that restoring inhibitory tone in the vHipp can alleviate dopamine system dysfunction and psychosis-like phenotypes in animal models used to study psychosis [26,27,28,29,30]. These data provide a strong rationale for targeting GABAergic neurotransmission in the hippocampus as a potential site of therapeutic intervention for the treatment of psychosis in patients with TLE.

One potential approach to normalize hippocampal activity may be to target γ-aminobutyric acid type A receptors containing the α5 subunit (α5GABA_A_Rs), which have a well-characterized expression profile, and are primarily expressed within the hippocampal formation [31,32]. Recent studies have demonstrated that positive allosteric modulators (PAMs), selective for α5GABA_A_Rs, can produce a host of beneficial effects, including antidepressant, anxiolytic, procognitive [33,34], and antipsychotic-like effects [28,35]. Specifically, α5-PAMs have been shown to restore aberrant dopamine neuron activity in two different animal models used to study psychosis [28,35]; however, these antipsychotic-like effects appear to be dependent on the extrasynaptic location of α5GABA_A_R (extrasynaptic versus synaptic [36].

Periods of high neural activity, like those seen during seizures, have been shown to shunt hippocampal α5GABA_A_Rs into the synapse in vitro [37], where an α5-PAM may be less effective [36]. For these reasons, we examined whether α5GABA_A_Rs translocate into the synapse in the pilocarpine model used to study TLE, and investigated if an α5-PAM, GL-II-73, was able to restore dopamine system function, using in vivo electrophysiology. GL-II-73 has been characterized as a positive allosteric modulator at the GABA_A_ receptor, with priority affinity and efficacy at α5GABA_A_Rs [33]. Specifically, Ki values at the α5 subunit were shown to be 6 times more selective compared to α2, 11 times more compared to α1, and 12.5 times more compared to α3. Efficacy at α5GABA_A_Rs also showed PAM activity, showing preferential α5-PAM activity compared to other α-subunits, most importantly compared to α1. We observed a significant increase in VTA dopamine neuron activity following recurrent seizures, which was reversed by intra-vHipp administration of the α5-PAM, GL-II-73. We also found that pilocarpine- and saline-treated rats did not differ in their total or synaptic expression of α5GABA_A_Rs. These results suggest that restoring hippocampal inhibitory tone with GL-II-73 may have clinical utility for the treatment of psychosis associated with epilepsy. Taken together, these data provide a strong rationale that targeting GABAergic neurotransmission in the hippocampus may be a potential site of therapeutic intervention for the treatment of psychosis in patients with TLE.

## 2. Results

### 2.1. Pilocarpine-Induced Seizures Increased VTA Dopamine Neuron Population Activity, Which Was Reversed by GL-II-73

To establish a model of TLE in rodents [38,39], we unilaterally injected pilocarpine directly into the vHipp. A representative brain slice showing the cannula track is depicted in Figure 1A. Consistent with previous reports [40], we found that rats that received an intra-vHipp injection of pilocarpine had a significantly higher Racine rating compared to saline-treated controls (Figure 1B, *t* test; *t* = 11.29, *p* < 0.0001).

Dopamine system function is known to be altered in patients with psychosis [24,25,42] and in rodent models used to study psychosis [21,43,44,45], as well as in pilocarpine-treated rats [40]. Consistent with previous findings, we observed a significant increase in dopamine neuron activity in pilocarpine-treated rats that received intra-vHipp vehicle (Figure 2A, *n* = 13 rats; 1.88  ±  0.10 cells per track; two-way ANOVA; F_Pilocarpine (1,33)_ = 15.370; *p* < 0.001; F_Drug (1,33)_ = 7.121; *p* = 0.012; Holm–Sidak; *t*  =  3.977, *p*  =  0.002) when compared to saline-treated rats that received intra-vHipp vehicle (*n* = 6 rats; 1.13  ±  0.13 cells per track). Interestingly, the pilocarpine-induced increase in dopamine neuron activity was completely reversed by the intra-vHipp administration of GL-II-73 (*n* = 12 rats; 1.30 ± 0.13 cells per track; Holm–Sidak; *t* = 3.800, *p* = 0.002), and had no effect in saline-treated rats (*n* = 6 rats; 1.00 ± 0.09 cells per track). Consistent with a previous observation [40], a subset of rats (4 of 17) did not develop a hyperdopaminergic phenotype (*n* = 4 rats; 1.05 ± 0.13 cells per track), were classified as non-responders (defined as >1.5 cells per track), and were not included in this dataset. As expected, no significant differences were observed in the average firing rate (Figure 2B; F_Pilocarpine(1,228)_ = 0.140; *p* = 0.709; F_Drug (1,228)_ = 0.039; *p* =  0.844) or bursting pattern (Figure 2C; F_Pilocarpine (1,228)_ < 0.001; *p* = 0.989; F_Drug (1,228)_ = 0.034; *p* =  0.853). It should be noted that while both male (circles) and female (squares) subjects were represented, we are underpowered to detect any sex differences.

### 2.2. Pilocarpine-Induced Seizures Did Not Alter α5GABA_A_R Expression within the vHipp

The localization of α5GABA_A_Rs (extrasynaptic vs. synaptic) plays a key role in the hippocampal modulation of dopamine neuron activity [36] and activity-dependent changes in α5GABA_A_R localization are known to occur [37]. Thus, we verified the expression and localization of α5GABA_A_Rs between saline and pilocarpine-treated rats. To examine synaptic localization, we measured gephyrin-associated α5GABA_A_Rs using coimmunoprecipitation. We found no differences in total α5 protein levels (Figure 3A, *t* test; *t* = 1.075 *p* = 0.314) or gephyrin-pulldown α5 protein levels (Figure 3B, *t* test; *t* = 1.631, *p* = 0.142) between pilocarpine-treated and saline-treated rats.

## 3. Discussion

Despite the fact that temporal lobe epilepsy (TLE) patients are 6–12 times more likely than the general population to suffer from psychosis, treatments for psychosis in this patient population remain inadequate [5,6]. Antipsychotic medications incompletely treat psychosis symptoms and are associated with substantial adverse effects [9]. Additionally, there is evidence to suggest that antipsychotics can lower seizure threshold and increase seizure incidence [7,8], and undergo potential pharmacokinetic and pharmacodynamic interactions with antiepileptic drugs [7]. Thus, there is an immediate need for novel therapeutics to treat psychosis in TLE patients.

Augmented hippocampal activity appears to be central to the pathology of both TLE and psychosis [16,21,46], with diminished GABAergic signaling within the hippocampus likely contributing to pathophysiology [20,47,48,49,50]. α5GABA_A_ receptors are inhibitory receptors highly enriched in the hippocampus [31,32]. Thus, we were interested to see if restoring hippocampal inhibition by administering an α5-PAM, GL-II-73, could reverse aberrant dopamine system function in a rodent model used to study TLE.

In this study, we induced spontaneous recurrent seizures by injecting the pro-epileptic compound, pilocarpine, directly into the vHipp [38,39,51]. The administration of pilocarpine, a muscarinic receptor agonist, is a well-established model used to study TLE in rodents [13,38,39]. Both systemic and intra-vHipp administration of pilocarpine can generate status epilepticus within minutes, and can produce spontaneous recurrent seizures after a latent period of several weeks, but intracranial administration is associated with a much lower mortality rate [51], hence our choice to use this route of administration. Inducing TLE with pro-epileptic compounds has been shown to result in behavioral and neurophysiological alterations associated with psychosis, including hypersensitivity to amphetamine [52] and increased VTA dopamine neuron activity [40]. Consistent with these studies, we observed increases in VTA dopamine neuron population activity, without changes to firing rate or burst firing (Figure 2). This is consistent with the broader body of literature demonstrating a link between hippocampal hyperactivity, aberrant dopamine neuron activity, and psychosis symptoms. Specifically, clinical studies have indicated that psychosis patients display hippocampal abnormalities [16,50,53]. Additionally, our laboratory and others have demonstrated that, in rodents, vHipp hyperactivity can profoundly augment VTA dopamine neuron activity via a multisynaptic circuit [21,22,23,54]. VTA population activity, rather than firing rate or burst firing, is consistently altered in animal models of psychosis, and is regulated by the hippocampus [21,22,54], whereas other aspects of VTA dopamine neuron activity, such as burst firing, are regulated by regions such as the laterodorsal tegmental and pedunculopontine tegmental nuclei [43,55,56]. Thus, we believe that the increased VTA dopamine neuron population activity observed in pilocarpine-treated rats was due to increased vHipp activity.

Consistent with previous reports [40], a subset of pilocarpine-treated rats (25%) did not display elevated dopamine neuron activity (<1.5 spontaneously active cells/track). This segregation of “responders” and “non-responders” was based on the clear bimodal distribution of dopamine neuron population activity observed in previous studies [57]. However, the presence of non-responders is consistent with clinical outcomes, in which only a portion of TLE patients develop psychosis [5,6]. While the origin of the non-responders is unknown, it has been suggested to be due to the use of anesthetic during dopamine neuron recordings, which acts on the GABAergic system [58,59], similar to many antiepileptic drugs [60], and may decrease hippocampal seizures [40].

Here we demonstrate that intra-vHipp pilocarpine can produce aberrant dopamine neuron activity, and that the intra-vHipp administration of GL-II-73 fully reverses dopamine system dysfunction caused by TLE induction. This is consistent with the growing body of literature demonstrating that attenuating activity in the vHipp using a variety of methods including pharmacological [23], surgical [30] or cell-based approaches [29] can restore aberrant dopamine system function in animal models used to study psychosis. Further, GL-II-73 and other α5-PAMs have shown promising therapeutic efficacy in multiple animal models used to study psychosis [28,35,61]. In the current study, we provide evidence that the therapeutic utility of GL-II-73 extends to epilepsy-related psychosis as well. This is an important discovery, as it has been shown that GABA_A_R expression may be altered in some rat models of TLE, and a reduction in α5GABA_A_ receptor expression may decrease drug efficacy [12,48,49,62]. Specifically, a downregulation of α5GABA_A_Rs in the CA1 and CA2 regions of the hippocampus has been observed in pilocarpine-treated rats [47]. While we did not observe any differences in α5GABA_A_R expression here, methodological differences could be responsible for this discrepancy. We took tissue following our electrophysiology experiments, which occurred mere days after the first spontaneous seizures appeared (which ranged from 2 to 6 weeks after pilocarpine administration), whereas others have observed differences in α5GABA_A_R in rats that had been treated with pilocarpine 3–4 months prior [47]. It is possible that the downregulation of α5GABA_A_Rs occurs as part of the more chronic stages of pilocarpine-induced epilepsy, and at our timepoint, changes are undetectable. While α5GABA_A_R expression may influence the efficacy of α5-PAMs, we have recently demonstrated that shifting α5GABA_A_Rs from the extrasynaptic space to the synapse abolishes the ability of GL-II-73 to modulate VTA dopamine neuron activity (McCoy et al., in submission). Since periods of high-frequency activity have been reported to shunt α5GABA_A_Rs to the synapse [37], we wanted to ensure that this does not occur in models of epilepsy. We found no differences in gephyrin-associated α5GABA_A_Rs or in total α5GABA_A_Rs, suggesting that the proportion of synaptic α5GABA_A_Rs also does not change with pilocarpine treatment. These results are consistent with the electrophysiology data, which demonstrate that intra-vHipp GL-II-73 can still restore aberrant VTA dopamine neuron activity, even in a model of hippocampal hyperactivity.

## 4. Materials and Methods

All experiments were performed in accordance with the guidelines outlined in the USPH Guide for the Care and Use of Laboratory Animals and were approved by the Institutional Animal Care and the Use Committees of UT Health San Antonio and the U.S. Department of Veterans Affairs.

### 4.1. Animals

Adult male (*n* = 11) and female (*n* = 11) Sprague Dawley rats (250–300 g) purchased from Envigo (Indianapolis, IN, USA) were used for all experiments. Rats were group housed until surgery and kept on a 12 h light/dark cycle. Food and water were provided *ad libitum*.

### 4.2. Seizure Induction

All survival procedures were performed under general anesthesia in a semi sterile environment. Rats were anesthetized with Fluriso (2–5% isoflurane, USP, with oxygen flow at 1 L/min) and placed in a stereotaxic frame (Kopf, Tujunga, CA, USA) with blunt, atraumatic ear bars. A core body temperature of 37 °C was maintained. For the intracranial administration of drugs, standard guide cannulas (23 gauge; PlasticsOne, Roanoke, VA, USA) with an internal cannula (30 gauge; PlasticsOne) extending 1 mm past the end of the guide were used. Rats were randomly assigned to receive either pilocarpine (2.4 mg/μL; 1 μL) or vehicle (1 μL; Dulbecco’s phosphate-buffered saline (PBS)) and unilaterally injected (∼0.5 μL per min) in the right vHipp (from bregma: AP −5.3, ML +5.0, from brain surface: DV −6.0 mm). The dose of pilocarpine was selected based on previously published work demonstrating that a majority of rats (~75%) given this dose display a hyperdopaminergic phenotype [40]. This route of administration (intra-vHipp) is preferred to systemic administration as it is associated with significantly lower mortality [51]. Even so, three rats that were administered pilocarpine died before electrophysiology could be performed. Rats were administered ketoprofen as an analgesic (5 mg/kg, s.c.) and monitored closely until conscious. Within 5 min of waking, the majority of rats displayed status epilepticus, which was quantified using a modified Racine motor scale [63] defined as the following: 0 = no abnormal behavior, 1 = mouth and facial spasms, 2 = head nodding/wet dog shakes, 3 = forelimb clonus, 4 = rearing, 5 = rearing and falling, racing, jumping. Animals exhibiting convulsions lasting more than 120 min were injected with midazolam (5 mg/kg; i.p.) to stop seizure activity. Rats were monitored by an experimenter daily in the morning starting at 8:00 am for two hours for the development of spontaneous seizures, which appeared 2–6 weeks following injection.

### 4.3. In Vivo Extracellular Dopamine Neuron Electrophysiology

As early as the day following the development of spontaneous seizures, but no later than 1 week after development, in vivo extracellular dopamine neuron electrophysiology was performed. Rats were anesthetized with 8% chloral hydrate (400 mg/kg, i.p.) and placed in a stereotaxic apparatus for the duration of the experiment, typically lasting no longer than three hours. GL-II-73 (100 ng/µL; 0.75 uL) or vehicle (85% H_2_O, 14% propylene glycol, 1% Tween80; 0.75 µL) were administered directly into the vHipp unilaterally such that each rat served as its own vehicle control. The vehicle side was always recorded first, and the hemispheres receiving each drug were counterbalanced. The dose of GL-II-73 was selected based on previous characterization [28,33,35]. Extracellular glass microelectrodes (impedance ~6–10 MΩ) were lowered into the VTA (from bregma: AP −5.3, ML ±0.6, and from brain surface: DV −6.5 to −9.0 mm) using a hydraulic micro-positioner (Model 640, Kopf). Spontaneously active dopamine neurons were identified using open filter settings (low-frequency cutoff: 30 Hz; high-frequency cutoff: 30 kHz) according to previously established electrophysiological criteria [64,65]. Three parameters of dopamine activity were measured and analyzed: (1) the number of dopamine neurons firing spontaneously per track (population activity), (2) average firing rate, and (3) proportion of action potentials occurring in bursts (defined as the incidence of spikes with <80 ms between them; termination of the burst is defined by >160 ms between spikes). Electrophysiological data were collected with commercially available software (LabChart software; Version 8 ADInstruments, Sydney, Australia).

### 4.4. Verification of Electrode and Cannula Placement

At the completion of all experiments, rats were sacrificed via chloral hydrate overdose and rapidly decapitated. Brains were extracted and cut down the midsagittal line into hemispheres. One hemisphere was fixed for at least 24 h (4% phosphate-buffered formaldehyde), cryoprotected (10% *w/v* sucrose in phosphate-buffered saline) until saturated and used to verify electrode (VTA) and cannula (vHipp) placement. Brains were coronally sectioned (25 µm) using a cryostat (Leica, Buffalo Grove, IL, USA). Sections containing electrode or cannula tracks were mounted onto gelatin-coated slides, stained with neutral red (0.1%) and thionin acetate (0.01%), and cover-slipped with DPX Mountant for histochemical confirmation within the VTA or vHipp (Figure 1A). The other hemisphere was stored and used for protein expression analysis.

### 4.5. Immunoprecipitation and Western Blot

After electrophysiology was completed, rats were sacrificed as described above and the vHipp from either the right or left hemisphere was dissected on ice. The vHipp was then homogenized with lysis buffer, centrifuged at 14,000× *g* for 2 min and the supernatants were aliquoted and stored at −80 °C until further use. The α5 subunit and its binding partners were immunoprecipitated using Dynabeads protein G magnetic beads (ThermoFisher, Waltham, MA, USA) according to the manufacturer’s protocol. Western blots were used to quantify α5 and gephyrin protein levels in both the immunoprecipitated samples and hippocampal homogenates. In short, proteins in the total lysate were separated in an SDS-PAGE gel followed by blotting onto a 0.2 µm nitrocellulose membrane using a trans-blot turbo transfer system (Bio-Rad, Hercules, CA, USA). Membranes were then incubated with a primary antibody against α5-GABA_A_Rs (1:1000), gephyrin (1:3000), or GAPDH (1:1000) overnight at 4 °C, and washed 3 times (10 min) with Tris-buffered saline with 0.1% Tween 20 (TBST) prior to incubation with a horseradish peroxidase-conjugated anti-mouse antibody (1:5000; for GAPDH only) or an anti-rabbit secondary antibody (1:5000) for 1 h at room temperature. Membranes were washed with TBST and incubated with Pierce™ ECL western blotting substrate (ThermoFisher) followed by exposure to X-ray film for detection. Antibodies were stripped using a commercially available stripping buffer (ThermoFisher). Densitometry analyses of immunoreactive bands were performed using the NIH Image J software version 1.53e from the scanned films. Densitometric arbitrary units obtained for α5GABA_A_R immunoreactive bands were normalized to GAPDH. For co-immunoprecipitation experiments, gephyrin bands were normalized to α5GABA_A_R levels.

### 4.6. Statistical Analysis

All data are represented as mean ± SEM with *n* values representing the number of rats per experimental group unless otherwise indicated. Electrophysiology data were analyzed using LabChart Pro Version 8 (ADInstruments, Colorado Springs, CO, USA). A two-way ANOVA of these data (factors: pilocarpine × drug) was performed and plotted using Prism software version 9.5.1 (GraphPad Software Inc.; San Diego, CA, USA). When significant main effects or interactions were detected the Holm–Sidak post-hoc test was used. Western blots and Racine ratings were analyzed by t-test and plotted in Prism. All tests were two-tailed, and significance was determined at *p* < 0.05.

### 4.7. Materials

The proprietary compound, GL-II-73, was supplied by the Centre for Addiction and Mental Health, Campbell Family Mental Health Research Institute (Toronto, ON, Canada). Pilocarpine hydrochloride (P6503), chloral hydrate (C8383), propylene glycol (P4347), DPX (06522) and Tween80 (P1754) were obtained from Sigma-Aldrich (St. Louis, MO, USA). Midazolam hydrochloride was purchased from Akorn Pharmaceuticals (Lake Forest, IL, USA). The α5 antibody was from R&D Systems (#PPS027; Minneapolis, MN USA) and the anti-rabbit-HRP secondary was from Cell signaling (#7074; Danvers, MA, USA). All other antibodies were from Abcam (Cambridge, UK), including gephryin (ab181382) and GAPDH (#9484). Dynabeads protein G magnetic beads (10003D), ECL substrate (32106), stripping buffer (21059), and coverslips (174942) were from ThermoFisher (Waltham, MA, USA). Gels (4569035) and membranes (1704158) were from Bio-Rad. All other chemicals or reagents were of either analytical or laboratory grade and purchased from various suppliers.

## 5. Conclusions

As with other subunit-selective compounds, the question arises of whether the biological effects that are observed are really mediated by the subunit for which the compound has some relative selectivity. Based on in vitro data, GL-II-73 displays significant selectivity for α5GABA_A_Rs, but we acknowledge that we did not formally demonstrate that the effects observed with GL-II-73 are truly mediated by α5GABA_A_Rs in this study. Future research could resolve this using transgenic approaches. Another interesting area of research would be to examine if the increased tonic inhibition cause by α5-PAMs may attenuate seizure activity in the hippocampus [66]. Thus, α5-PAMs may provide relief for both epileptic and psychotic symptoms associated with TLE. Future studies should examine the potential antiseizure effects of α5-PAMs, as GL-II-73 may be beneficial for treating multiple pathologies associated with TLE [33,67]. As there are currently no viable treatment options for managing psychosis in epilepsy patients, this study provides promising preclinical evidence that GL-II-73 may be able to provide relief for those suffering from TLE with comorbid psychosis.

## Figures and Tables

**Figure 1 ijms-24-11588-f001:**
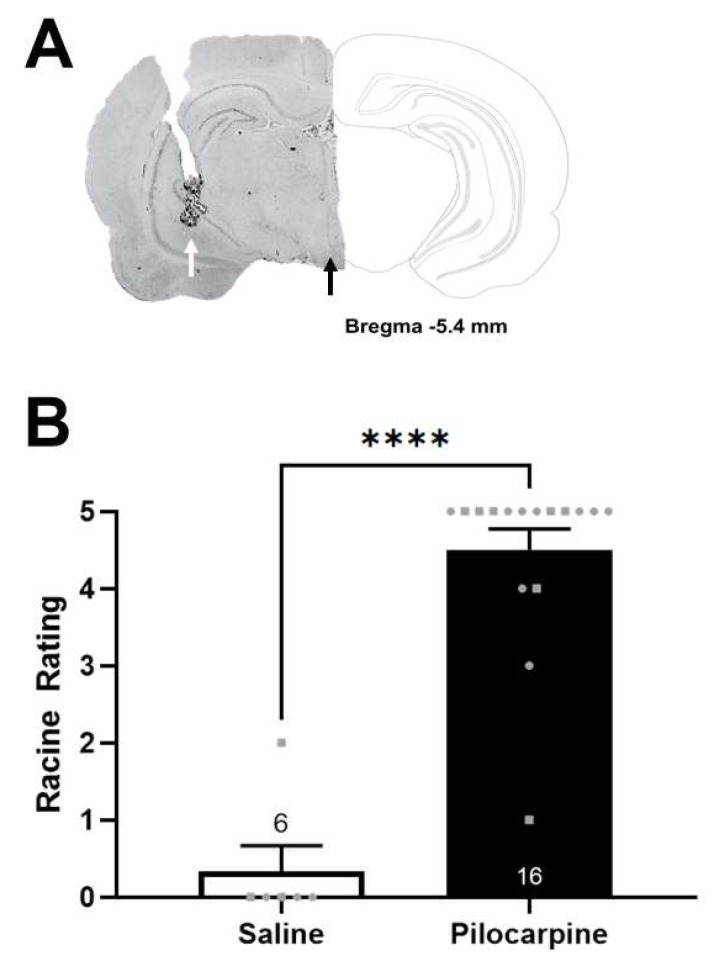
Verification of placements and response to pilocarpine. (**A**, **Left**) Representative brain slice containing an electrode track in the ventral tegmental area (VTA; black arrow) and cannula track in the ventral hippocampus (vHipp; white arrow) with corresponding schematic of the brain section (**right**, −5.40 mm posterior to bregma, [41]). (**B**) Pilocarpine-treated rats exhibited significantly higher average Racine ratings compared to saline-treated controls, **** *p* < 0.0001, *n* = 6–16/group; circles denote male subjects and squares denote females.

**Figure 2 ijms-24-11588-f002:**
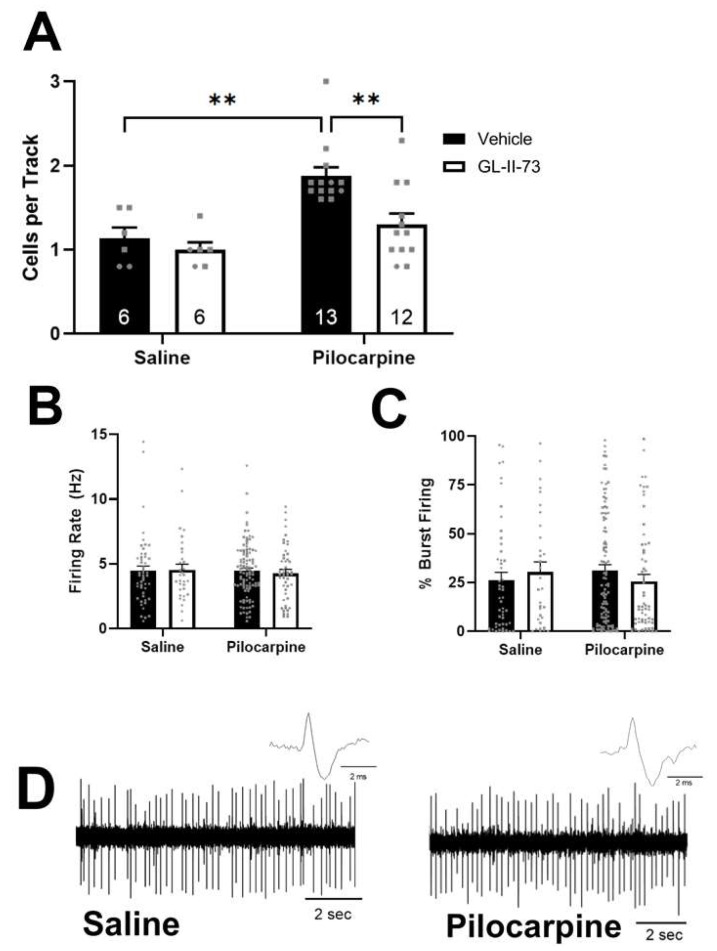
GL-II-73 reverses pilocarpine-induced increases in VTA dopamine neuron activity. (**A**) Intra-ventral hippocampal (vHipp) pilocarpine significantly increased the number of spontaneously active dopamine cells per track (population activity), which was reversed by intra-vHipp administration of GL-II-73 (100 ng/μL; 0.75 μL). Average firing rate (**B**) and percent burst firing (**C**) were not changed by pilocarpine or drug treatment. (**D**) Representative trace of dopamine neuron firing and action potential for saline-treated (**left**) and pilocarpine-treated (**right**) rats. ** *p* < 0.005, *n* = 6–13/group; circles denote male subjects and squares denote females.

**Figure 3 ijms-24-11588-f003:**
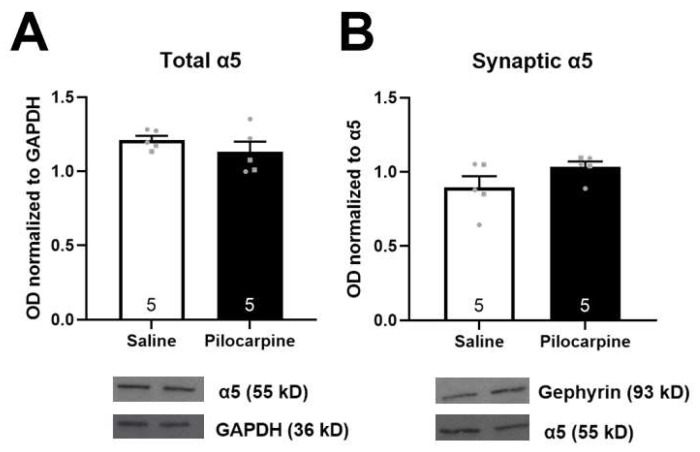
Pilocarpine does not alter total or synaptic α5 protein. There were no differences in total α5 protein expression; (**A**) co-immunoprecipitation of α5 and gephyrin was used to measure synaptic α5 protein (**B**) between saline- and pilocarpine-treated rats. Representative bands from saline- (**left**) and pilocarpine-treated rats (**right**) are below graphs. *n* = 5 rats/group, circles denote male subjects and squares denote females.

## Data Availability

The data presented in this study are available on request from the corresponding author.

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
