# Peer review of "GL-II-73, a Positive Allosteric Modulator of α5GABAA Receptors, Reverses Dopamine System Dysfunction Associated with Pilocarpine-Induced Temporal Lobe Epilepsy"

_ijms, 2023, doi:10.3390/ijms241411588_

Round 1

Reviewer 1 Report

The authors McCoy et al. have studied the effect of alpha5GABAAR-selective positive allosteric modulator (GL-II-73) on pilocarpine-induced temporal lobe epilepsy. They show that ventral hippocampal administration of GL-II-73 reverses the pilocarpine-induced increased ventral tegmental area dopamine neuron activity. Pilocarpine treatment did not affect alpha5GABAAR expression or localization.

The article is clear and very well written. 12 of the cited 58 references are from the last 5 years. No excessive self-citation is detected.

The experiments are expertly performed. The methods are written in details making it possible to be reproduced by other groups. The figures show clearly the data.

Comments:

In the methods the authors state that the densitometric arbitrary units obtained for alpha5GABAAR immunoreactive bands were normalized to GAPDH and expressed as percent of control. However, there are no percentage values in Fig. 3. The values in both 3A and 3B are close to 1.0 where control (saline) could therefore be 1.0 (100%), but saline is either over (A) or below (B) 1.0. Was there an untreated control that serves a control for saline and pilocarpine treatment? Could you please explain this.

Author Response

Apologies for the miscommunication. The arbitrary densitometric units found for alpha5 were normalized to GAPDH, but this is expressed as a number value, not a percentage. This was the case for both saline and pilocarpine-treated samples. This is why the saline values are not "set" at 1, or 100%. 

We have edited the manuscript so the methods now read "densitometric arbitrary units obtained for α5GABAAR immunoreactive bands were normalized to GAPDH." and hope this is sufficient to address the reviewer's concern. 

Reviewer 2 Report

This manuscript describes that in a model of pilocarpine-induced temporal lobe epilepsy (TLE) GL-II-73, administered into the ventral hippocampus, reverses the increase in VTA dopamine neuron activity. As patients with TLE have an increased risk to develop psychosis which is associated with increased activity in VTA dopamine neurons and as the use of antipsychotics is problematic for a variety of reasons, this result is of high potential significance. There are only three figures in the manuscript. One of them shows the main result just mentioned, and the two others show what are basically control experiments.  Thus, in terms of amount and depth of experimental data, this manuscript  is at the lower end of the spectrum, compared to many other papers in this journal. However, the presented data are convincing and the manuscript is very well written. A few items should be addressed in a revised version.

Major points

11)    The authors do not provide information on the selectivity of GL-II-73 for a5GABAARs. While affinity and efficacy data at different receptor subtypes are published elsewhere, the authors should provide a brief summary of these data in the Introduction.

22)    As with other GABAAR subtype-selective compounds, the question arises whether the biological effects that are observed are really mediated by the GABAAR subunit for which the compound has some relative selectivity. The authors should discuss that while GL-II-73 based on in vitro data displays significant selectivity for a5GABAARs, they formally have not demonstrated that the effects that they are observing with GL-II-73 are truly mediated by a5GABAARs.

Minor points

33)    Line 7: Eliminate the “and” between “Lodge” and “Ph.D.”

44)    Line 26. “Gamma”. Use the Greek letter or “gamma”.

55)    Line 29. “demonstrated”. Consider changing to “demonstrate”.

66)    Line 59. References “(Brooks-Kayal”: The “et al., 1998” is missing.

77)    Line 59. I could not find the references “DeDeyn P., 1990; Tasker, J. & Dudek 1990)” in the reference list. Please make sure that all references that are cited in the text also appear in the reference list.

88)    Lines 62/63. I could not find the reference “(Heckers & Konradi, 2002)” in the reference list.  Please add this reference.

99)    Line 89. The citation of unpublished results as “in submission” is somewhat unusual. Why do not deposit a preprint with bioRxiv and cite it in the reference list ?

110) Line 217. “Davers, MA, USA”. Please correct to “Danvers, MA, USA”.

111) Lines 250/251: “It should be noted that while both male (circles) and females (squares) subjects were represented…” and line 260: “circles denote male subjects and square denote females”. It is almost impossible to distinguish between circles and squares in Fig 2. The authors might consider using larger symbols or including the information how many male and female animals were used in each group in the Materials and Methods section.

Author Response

The authors would like to thank the reviewer for their keen eye and helpful comments. Please see attached file for line-by-line responses. 
